# SAG: STYLE-ALIGNED ARTICLE GENERATION VIA MODEL COLLABORATION

## ABSTRACT

Large language models (LLMs) have increased the demand for personalized and stylish content generation. However, closed-source models like GPT-4 present limitations in optimization opportunities, while the substantial training costs and inflexibility of open-source alternatives, such as Qwen-72B, pose considerable challenges. Conversely, small language models (SLMs) struggle with understanding complex instructions and transferring learned capabilities to new contexts, often exhibiting more pronounced limitations. In this paper, we present a novel collaborative training framework that leverages the strengths of both LLMs and SLMs for style article generation, surpassing the performance of either model alone. We freeze the LLMs to harness their robust instruction-following capabilities and subsequently apply supervised fine-tuning on the SLM using style-specific data. Additionally, we introduce a self-improvement method to enhance style consistency. Our new benchmark, NoteBench, thoroughly evaluates style-aligned generation. Extensive experiments show that our approach achieves state-of-the-art performance, with improvements of 0.78 in ROUGE-L and 0.55 in BLEU-4 scores compared to GPT-4, while also maintaining a low hallucination rate in terms of factual accuracy and faithfulness.

## 1 INTRODUCTION

Stylish article generation is an emerging and significant research topic in AI-generated content (AIGC). A critical challenge in this domain is the comprehensive understanding of various writing styles, including tone, vocabulary, and structural nuances. This complexity poses difficulties for models, making it difficult to effectively replicate a specific style while ensuring that the generated content remains coherent and contextually relevant. Previous works primarily focus on stylish article transfer while lacking attention to content generation. Specifically, studies such as Horvitz et al. (2024b) and Khan et al. (2024) often transform approximately dozens of words of text from one style to another without creating any new content. Recently, large language models (LLMs) pre-trained on vast amounts of data have demonstrated significant potential in various generation tasks, such as writing (Coenen et al., 2021; Lee et al., 2022; Chung et al., 2022), mathematics (Yang et al., 2024; Shao et al., 2024), and coding (Team et al., 2024; DeepSeek-AI et al., 2024; Hui et al., 2024). These models excel at following specific instructions and executing generative tasks, offering promising solutions for stylish article generation.

Existing works that employ LLMs in generating stylish articles can be categorized into two approaches: training-free and training-based methods. The training-free approach (*e.g.*, prompt engineering, and in-context learning) typically involves crafting specific prompts or examples to guide LLMs in adopting a particular style while generating articles. This method heavily relies on the world knowledge and comprehension capabilities embedded within the LLMs. Moreover, these models, such as GPT, LLaMA, and Qwen, are primarily trained on general data, which may limit their effectiveness in stylish generation tasks. Conversely, the training-based approach enhances the LLMs' stylistic generation capabilities through supervised fine-tuning (SFT) on targeted stylish data. However, this adjustment to a pre-trained LLM often results in catastrophic forgetting (Dou et al., 2023; Lin et al., 2024), diminishing the model's instruction-following capabilities and leading to a loss of world knowledge. Additionally, applying SFT to LLMs presents challenges: for instance, some models are closed-source, and others are too large to fine-tune effectively. Smaller language models (SLMs) could serve as a potential alternative, but they typically exhibit inferior ca-

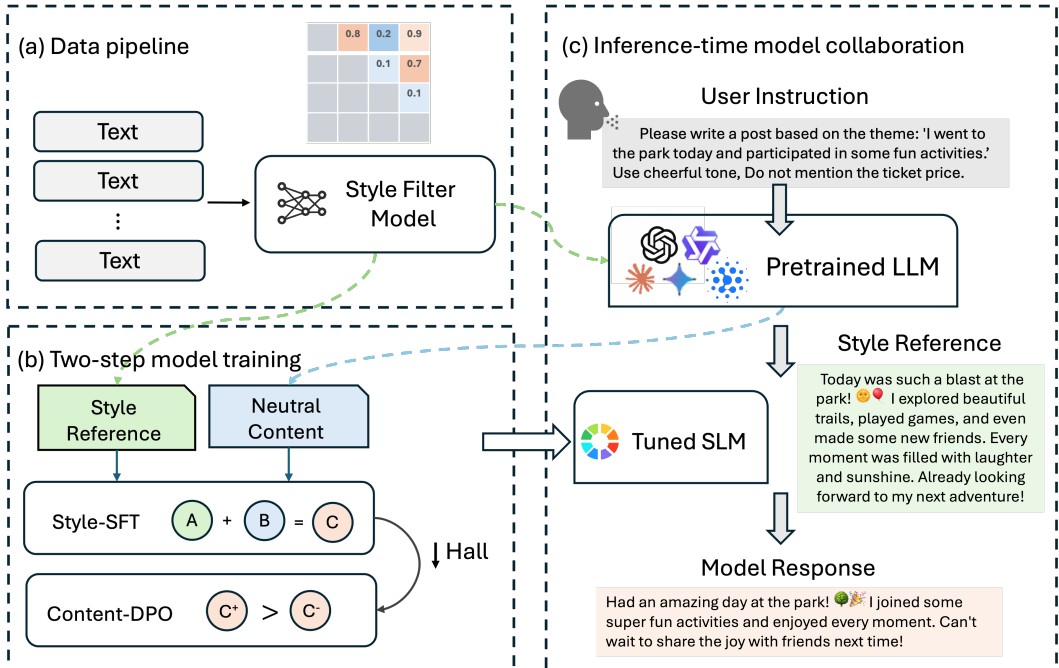

Figure 1: The framework of the collaborative training of style-aligned article generation: User instruction is initially processed by a pre-trained LLM such as GPT4 to formulate neutral content. Subsequently, the style reference content, user instruction, and neutral content are fed to an SLM to generate stylish content.

pabilities in understanding, integration, and hallucination control compared to LLMs. This situation highlights the difficulties of balancing style-following capabilities with effective content generation.

We propose a novel style-aligned generation framework that incorporates collaborative training of small language models (SLM) and large language models (LLM). In this framework, we first freeze the LLM and then train the SLM. The frozen LLM, which possesses instruction-following abilities and world knowledge, serves as an interface to process user instructions for the SLM. The training process for the SLM consists of two stages: style supervised fine-tuning (S-SFT) and content direct preference optimization (C-DPO). In the S-SFT stage, we filter style-consistent data to fine-tune the SLM, thereby enhancing its capacity to generate stylish articles. To improve style consistency, we introduce a self-improved approach to filter out training pairs with low style similarity. In the C-DPO stage, we curate content preference data to further mitigate the hallucination issues in the SLM. Finally, given a user instruction, the LLM generates a neutral article, and then the SLM injects the article with a specific style, achieving superior performance in both contextual and stylish refinement.

We introduce a new benchmark named NoteBench to evaluate performance. Specifically, we define a style imitation task in which the content summary and stylish references are treated as inputs, while the stylish content serves as the output. We employ Rouge (Lin) and BLEU (Papineni et al.) metrics to assess style consistency and use GPT-4 to evaluate the degree of hallucination regarding factual accuracy and faithfulness. Extensive experiments demonstrate that our collaborative training approach for SLMs and LLMs surpasses existing state-of-the-art (SOTA) LLMs when compared to prompt engineering and vanilla SFT on the same SLM. For instance, we achieve improvements of 0.78 in ROUGE-L and 0.55 in BLEU-4 compared to GPT-4, and 2.86 in ROUGE-L and 1.92 in BLEU-4 with vanilla SFT. Additionally, our method maintains a low hallucination rate in comparison to GPT-4 while significantly surpassing vanilla SFT, with a decrease of 29.1%.

We summarize our contributions as follows:

1. We propose a novel collaborative training framework that synergizes the strengths of both large and small language models, achieving superior performance compared to the use of a single model in isolation.

2. We define the style imitation task, paving the way for an innovative stylish article generation approach in large unlabeled or semi-labeled datasets. Additionally, we introduce NoteBench, a benchmark specifically designed for this task, enabling the research community to assess the composition and imitation capabilities of language models, thereby facilitating further advancements in this area.

3. We achieve state-of-the-art performance through our collaborative training approach for small and large language models, demonstrating significant improvements in stylish article generation and hallucination mitigation.

## 2 RELATED WORK

One relevant research topic in stylish article generation is text style transfer, which aims to transfer an article from a source style into a target style while preserving the original meaning. Unsupervised style transfer reconstructs texts and style representations from non-parallel examples (Krishna et al., 2020; Riley et al., 2021; Horvitz et al., 2024a). Krishna et al. (2020) first employs paraphrasing to neutralize the style article text and trains the model to re-stylize it. Riley et al. (2021) propose a style transfer approach that enables a single model to generate various styles by conditioning on different target styles. Several studies also leverage LLMs for style transfer through zero-shot or few-shot in-context learning (Patel et al., 2023; Reif et al., 2022). However, they primarily focus on converting pre-existing sentences or passages, overlooking the collaboration between LLMs and SLMs, as well as the challenges posed by information expansion in the creation of new content.

Stylish article generation is also prevalent in role-playing, where LLMs are utilized to portray specified characters by imitating their tones and behaviors. RoleLLM (Wang et al., b) first proposes a benchmark for aligning LLMs with specific character language styles, demonstrating that injecting style information into LLMs using auto-annotated data is effective. CharacterLLM (Shao et al.) further advances the concept of personified LLMs in more practical scenarios. By editing character profiles, these models are trained to provide the experiences and knowledge inherent to the characters. DITTO (Lu et al.) proposes a self-alignment method that uses self-generated datasets to augment its role-playing capabilities. Several works also focus on evaluating performance in role-playing (Wang et al., 2024; Jiang et al.). However, they primarily enhance persona-related performance, often centering on celebrity personas and overlooking the nuanced expressions and writing styles of real-world users.

Alignment technology is widely used to adapt the base LLM to better align with human intentions, including human values (Gabriel, 2020), domain knowledge (Zhang et al., 2024), and task-solving abilities (Sun et al., 2024). Preference datasets are obtained through human feedback (Ouyang et al., 2024) or LLM annotations (Wang et al., 2023). Training methodologies, such as Proximal Policy Optimization (PPO) (Schulman et al., 2017), Direct Preference Optimization (DPO) (Rafailov et al., 2024), and Prospect Theory Optimization (KTO) (Ethayarajh et al., 2024) use human preference data to improve model performance. In this task, we aim to align the models with stylish text generation, specifically by composing content from relatively short summaries while maintaining factuality and faithfulness.

## 3 STYLE-ALIGNED ARTICLE GENERATION

### 3.1 TASK DEFINITION

To emphasize stylish content generation, we streamline the general AI content generation process and define the style imitation task. We simplify the process of topic conception and related information collection to the summary part, which contains sufficient topics and information for generating an article. Nevertheless, the summary is relatively less stylish and structured, enabling the model to generate a more characterized article through the style reference part. The task emphasizes the content fidelity of the summary and style consistency with the reference text, allowing us to focus on generating stylistically consistent text without delving into other complex AI content generation

techniques. Notably, the style reference text is provided by the complete content rather than a collection of explicitly defined style types, which requires the model to implicitly mimic the style from the reference text.

## 3.2 STYLE DATA CURATION

**Data Curation** We collect articles from open-source forums, including product feature introductions and user experience sharing. These articles span a variety of topics such as fashion, daily life, and travel, originating from users of diverse backgrounds. We cluster the articles by user and sort them by publication date. This ensures that only the content previously posted by each user is referenced, maintaining chronological consistency.

**Self-improvement Style Filter** The content style of the same user may vary, leading to suboptimal training. To enhance consistency, we introduce a self-improvement method as shown in Algorithm. 1, inspired by recent works (Yang et al.; Wang et al., a; Chen et al.). We construct a style dataset with positive pairs from the same user and negative pairs from different users, which we use to train a style filter model. We then filter out pairs in the training set with style similarity above a specified threshold. To ensure adequate stylistic information, we exclude articles with minimal word counts. We utilize a BERT-like text-to-embedding model to measure style consistency through the cosine similarity of generated paragraph vectors. We employ an in-batch negative loss (Gillick et al.; Yih et al.):

$$\mathcal{L} = -\frac{1}{n} \sum_{i=1}^{n} \log \frac{\exp(sim(a_i, p_i))}{\sum_{k=1}^{n} \exp(sim(a_i, p_k)) + \sum_{k=1}^{m} \exp(sim(a_i, n_k))} \tag{1}$$

where $n$ is the number of positive anchor-positive pairs, $m$ is the number of hard negatives for each anchor-positive pair, $sim(a, p)$ is a cosine similarity function between anchor $a$ and positive sample $p$, $a_i$ represents the anchor in the $i$-th pair, $p_i$ represents the positive sample corresponding to anchor $a_i$, and $n_k$ represents a hard negative sample for anchor-positive pairs.

We employ the style filter model to compute the similarity scores within each article user's submissions. Articles posted by the same user are pairwise associated with a similarity score to form a scoring matrix. To ensure that only previously posted articles are used as style references, we select only the upper triangular portion of the matrix and retain pairs exceeding a certain threshold. Finally, we construct nearly one million articles with over 400,000 unique users. Tab. 2 provides the detailed statistics of the constructed stylish article dataset, alongside comparisons with other character datasets.

---

**Algorithm 1:** Self-Improvement for Style Consistency

---

**Data:** Character Data Base $\mathcal{D}_C$, Seed model $\mathcal{M}$
**Result:** Style-Consistent Dataset $\mathcal{D}_R$, Style Filter $\mathcal{M}_R$
// Model Training
$\mathcal{M}_R = \text{Train}(\mathcal{D}_C)$
$\mathcal{D}_R = []$
**for** *user_collections in* $|\mathcal{D}_C|$ **do**
    **for** *article_pair in user_collections* **do**
        // Filter out unrelated articles
        **if** $\mathcal{M}_R$ *(article_pair)* > *threshold* **then**
            $\mathcal{D}_R$.append($q$)

---

## 3.3 COLLABORATIVE TRAINING

We introduce a collaborative training framework that combines a frozen LLM for processing raw user instructions with an SLM for generating stylish articles. This approach allows the SLM to produce results that are both contextually appropriate and stylistically superior, thereby surpassing the performance of either model when used individually.

**Simulating Intention with LLM.** We adopt an inverse generation approach (Riley et al., 2021) to align user-generated content with user intentions. Specifically, we instruct the SOTA LLM to extract a summary and generate neutral text that simulates user intention. This simulates a scenario where users instruct the LLM to create content while seeking additional stylish refinement. For summary extraction, we instruct the LLM to include all key information from the original content, such as entities, time, and price. For neutral text generation, akin to paraphrase generation (Krishna et al., 2020), we instruct the LLM to retain essential details while minimizing stylish expressions—avoiding specialized vocabulary, emojis, and uncommon punctuation. Finally, the extracted summaries, generated neutral text, and original content are organized into a training dataset. The detailed prompts and examples are provided in Appendix A.

**Imitating Style with SLM.** We conduct style supervised fin-tuning (S-SFT) to generate stylish content. Specifically, we incorporate both neutral text and style reference articles as inputs. This combination offers both style and content guidance, facilitating more effective stylish content generation. By using neutral text as an input, we can ensure that essential information is preserved while allowing the stylish elements to be infused. To further enhance this process, we include the original summary to compensate for any potential loss of information that may occur during the LLM's processing. This ensures that critical content details are not overlooked, allowing the SLM to generate outputs that maintain both the richness of the original information and the desired stylish features. Ultimately, this approach promotes a balance between informative content and stylistic refinement, ensuring high-quality results that meet user expectations. We use the cross-entropy loss on the generated tokens, as shown below:

$$\mathcal{L}(\theta) = -\sum_{t=1}^{T} \log P_\theta(y_t | y_{<t}, S, N, R) \tag{2}$$

where $\theta$ are the parameters of the SLM, $y_t$ is the $t$-th token of the output stylized text, $y_{<t}$ represents all tokens in the output sequence before time step $t$, $S = \{s_1, s_2, \ldots, s_{T_s}\}$ is the summary, $N = \{n_1, n_2, \ldots, n_{T_n}\}$ is the input sequence of neutral text, $R = \{r_1, r_2, \ldots, r_{T_r}\}$ is the style reference, and $P_\theta(y_t | y_{<t}, S, N, R)$ is the probability of the token $y_t$ given the previous tokens $y_{<t}$ and other contexts.

Through S-SFT, we effectively train the SLM to recognize and replicate various stylish traits, such as tone, vocabulary choice, and sentence structure, that align with user intention. The training dataset is curated to include diverse examples, showcasing a range of styles and contexts, which enhances the model's adaptability. As a result, the generated content will not only be contextually relevant but also exhibit a polished and engaging style, ultimately improving user satisfaction.

**Mitigating Hallucination with DPO.** Given that the SLM is more prone to generating content with hallucinations, we further align it using a preference dataset from the previous stage. Specifically, we employ a held-out alignment dataset, sourced similarly to the training dataset. The fine-tuned model in S-SFT is then used to infer on this alignment dataset. We leverage the LLM to correct the hallucination segments of the generated content while preserving other parts to maintain stylistic integrity. Since the verified content is typically shorter than the original, which may lead to reward hacking (Pan et al., 2024), we not only verify the hallucinated content but also instruct the LLM to incorporate additional information from the reference content.

During the training of the policy model, we consistently use the verified content as the chosen sample and the original generated content as the rejected sample, as shown below:

$$\mathcal{L}_{\text{DPO}}(\pi_\theta; \pi_{\text{ref}}) = -\mathbb{E}_{(x, y_w, y_l) \sim \mathcal{D}} \left[ \log \sigma \left( \beta \log \frac{\pi_\theta(y_w \mid x)}{\pi_{\text{ref}}(y_w \mid x)} - \beta \log \frac{\pi_\theta(y_l \mid x)}{\pi_{\text{ref}}(y_l \mid x)} \right) \right]. \tag{3}$$

where $\hat{r}_\theta(x, y) = \beta \log \frac{\pi_\theta(y|x)}{\pi_{\text{ref}}(y|x)}$ is the reward implicitly defined by the language model $\pi_\theta$ and reference model $\pi_{\text{ref}}$. The examples are weighed by how much higher the implicit reward model $\hat{r}_\theta$ rates the articles with wrong information, scaled by $\beta$, i.e, how incorrectly the implicit reward model orders the completed articles, accounting for the strength of the KL constraint.

# 4 EXPERIMENT

## 4.1 EXPERIMENTAL SETTINGS

**Base LLMs.** We select SOTA closed-source and open-source models as the base LLMs. Specifically, we choose Qwen for open-sourced models and choose GPT-4 (Achiam et al., 2023), GLM-4 (GLM et al., 2024), Claude-3 and Gemini-Pro for closed-source models. These models produce the neutral style content for SAG. We report the performance of style consistency and hallucination on these models in Tab. 1 as the baseline.

Table 1: Performance of style consistency and hallucination on SOTA closed-source and open-source LLMs.

| LLM | Open-Source | Style Consistency ↑ | | | | Hallucination ↓ | |
| --- | --- | --- | --- | --- | --- | --- | --- |
| | | Rouge-1 | Rouge-2 | Rouge-L | BLEU-4 | Factual | Faithful |
| Qwen | ✓ | 31.28 | 11.24 | 20.83 | 11.20 | 10.25 | 24.76 |
| GPT-4 | ✗ | 35.55 | 14.34 | 23.61 | 14.28 | 11.75 | 32.10 |
| Claude-3 | ✗ | 31.68 | 10.97 | 18.37 | 9.90 | 17.08 | 38.66 |
| Gemini-Pro | ✗ | 32.23 | 12.69 | 20.95 | 11.92 | 22.54 | 45.49 |
| GLM-4 | ✗ | 37.27 | 15.76 | 25.61 | 14.09 | 5.05 | 12.43 |

**Implementation Details.** We select GTE-large (Li et al., b) as the initialized style filter model. The model is trained on a stylish article dataset for 3 epochs with a batch size of 16, a learning rate of 2e-5, and a warm-up period for the first 5% of training samples. We select Qwen-1.8B and Qwen-7B as the SLMs. The training process consists of two steps: S-SFT and C-DPO. In the S-SFT stage, the batch size is set to 16 and the maximum sequence length is set to 2048. We adopt a learning rate of 1e-5 with cosine decay. The model is trained for 5 epochs, and the first 1% of training samples are warmed up. In the C-DPO stage, the learning rate is reduced to 1e-6, and the model is trained for 1 epoch while keeping all other hyperparameters the same as in the S-SFT stage.

**Training Datasets.** We utilize multiple datasets across various stages of our training process, specifically during style filtering, style supervised fine-tuning (S-SFT), and collaborative desired preference optimization (C-DPO). As detailed in Tab. 2, these datasets are carefully curated to ensure a comprehensive representation of styles and content types.

Table 2: Comparison of character datasets, #Character indicates the number of characters or the number of real-world article users, and #Samples indicates the total number of generated articles or contents.

| Model | #Character | #Samples |
| --- | --- | --- |
| Character-LLM (Shao et al.) | 9 | 1,900,800 |
| ChatHaruhi (Li et al., a) | 32 | 54,726 |
| RoleLLM (Wang et al., b) | 100 | 140,726 |
| CharacterGLM (Zhou et al.) | 250 | 1034 |
| DITTO (Lu et al.) | 4,002 | 36,662 |
| *Ours* | 429,791 | 943,685 |

**Evaluation Benchmarks** We use NoteBench to evaluate the performance of different methods on the style imitation task, which consists of 249 users and 732 articles. To assess style consistency, we employ ROUGE and BLEU as evaluation metrics. BLEU assesses the precision of the generated text by focusing on the overlap of n-grams between the generated output and reference texts. In contrast, ROUGE measures recall by evaluating the similarity between the model's responses and the target answers, identifying the proportion of overlapping words or phrases. To control hallucinations, we prompt GPT-4o to evaluate the hallucination rate of generated content, focusing on two key aspects: **Factual Hallucinations**: These occur when there is a discrepancy between the generated content and real-world facts, resulting in inaccuracies or fabrications; **Faithful Hallucinations**: These arise when the generated content diverges from user instructions or the context provided by the input, as well as when there is a lack of self-consistency within the content.

## 4.2 MAIN RESULTS

Table 3: Performance on style-consistency and hallucination. We report the collaboration of diverse SLMs and LLMs. red denotes a superior performance. green denotes a inferior performance.

| SLM | LLM | Style Consistency ↑ | | | | Hallucination ↓ | |
|---|---|---|---|---|---|---|---|
| | | Rouge-1 | Rouge-2 | Rouge-L | BLEU-4 | Factual | Faithful |
| 1.8B | Qwen | $35.20_{+3.92}$ | $14.39_{+3.15}$ | $22.12_{+1.29}$ | $11.85_{+0.65}$ | $9.02_{-1.23}$ | $25.55_{+0.79}$ |
| | GPT-4 | $35.88_{+0.33}$ | $14.92_{+0.58}$ | $22.88_{-0.73}$ | $13.61_{-0.67}$ | $16.39_{+4.64}$ | $33.47_{+1.37}$ |
| | Claude-3 | $34.39_{+2.71}$ | $13.66_{+2.69}$ | $21.02_{+2.65}$ | $11.22_{+1.32}$ | $15.85_{-1.23}$ | $31.97_{-6.69}$ |
| | Gemini-Pro | $33.78_{+1.55}$ | $13.37_{+0.68}$ | $21.66_{+0.71}$ | $11.88_{-0.04}$ | $17.90_{-4.64}$ | $38.93_{-6.56}$ |
| | GLM-4 | $38.34_{+1.07}$ | $16.84_{+1.08}$ | $25.43_{-0.18}$ | $13.63_{-0.46}$ | $5.60_{+0.55}$ | $12.43_{-0.02}$ |
| 7B | Qwen | $36.35_{+5.07}$ | $15.25_{+4.01}$ | $23.81_{+2.98}$ | $13.09_{+1.89}$ | $8.33_{-1.92}$ | $25.41_{-0.65}$ |
| | GPT-4 | $36.85_{+1.30}$ | $15.50_{+1.16}$ | $24.39_{+0.78}$ | $14.83_{+0.55}$ | $12.16_{+0.41}$ | $30.60_{-1.50}$ |
| | Claude-3 | $35.59_{+3.91}$ | $14.71_{+3.74}$ | $22.69_{+4.32}$ | $11.60_{+1.70}$ | $14.48_{-2.60}$ | $32.92_{-5.74}$ |
| | Gemini-Pro | $34.99_{+2.76}$ | $14.13_{+1.44}$ | $23.11_{+2.16}$ | $12.93_{+1.01}$ | $15.16_{-7.38}$ | $37.02_{-8.47}$ |
| | GLM-4 | $39.14_{+1.87}$ | $17.39_{+1.63}$ | $26.84_{+1.23}$ | $13.64_{-0.45}$ | $3.83_{-1.22}$ | $11.07_{-1.36}$ |

**Style Consistency.** We evaluated the composition and imitation abilities through several experiments, with the results compared with different LLMs presented in Tab. 3. SAG shows consistent quality improvement for both 1.8B and 7B models when using Qwen as the collaborating LLM. Furthermore, although trained using neutral content generated by Qwen, our SAG framework enhances style consistency across various other LLMs, achieving an average improvement of 0.55 in BLEU-4 and 1.52 in ROUGE-L. Additionally, we observe the "strong model advantage", where stronger LLMs yield better overall results.

**Hallucination Mitigation.** we also compare the hallucination rates among different methods in Tab. 3 and Tab. 4. Our SAG method shows promising performance in hallucination control. After training in the C-DPO stage, the SLM achieves comparable or even slightly better performance compared with LLMs, with an average of 2.91% rate decrease and a significantly better performance compared with SLM only trained in the S-SFT stage. The results show our SAG method can effectively mitigate the hallucination during content generation.

**Comparison of Collaborative Training.** We employ style reference texts and neutral contents generated by the LLM for the style imitation task, termed the S-SFT phase. Then we apply preferred pairs to mitigate hallucinations, referred to as the C-DPO phase. To evaluate the impact of these processes, we conduct experiments using both the 1.8B and 7B models, as shown in Tab. 4. The average improvement after alignment in BLEU-4 is 0.26 and Rouge-L is 0.81, indicating that the alignment process effectively reduces hallucinations while preserving the text style.

## 4.3 ABLATION STUDY

**Effect of SAG.** We compare the performance of our SAG method with two commonly adopted methods: vanilla SFT and the Text Style Transfer method (TST). For a fair comparison, we only use the S-SFT stage model in our SAG method. as shown in Tab. 5 and Tab. 6. Our proposed SAG approach incorporates the original user instruction, style reference, and the article generated by Qwen-72B-Chat. Additionally, we fine-tune two other models: one using a combination of user instruction and style reference (Vanilla SFT) and the other using the article generated by Qwen-72B-Chat along with the style reference for comparison (TST). For the LLM-involved methods TST and SAG, we employ a large language model to generate content based on user instruction and style reference. We observe that the LLM-involved method outperforms the Vanilla SFT method, likely because the task is partly solved by the large language model and is simpler for the smaller model, making it easier for the smaller model to learn effectively. Furthermore, the SAG method achieves the best results among the three approaches, possibly due to information loss during the composition of the LLM. Thus, the smaller model benefits from access to both the LLM-generated article and the user instruction.

Table 4: Performance of S-SFT and C-DPO.

| LLM | SLM | Stage | | Style Consistency ↑ | | | | Hallucination ↓ | |
|---|---|---|---|---|---|---|---|---|---|
| | | S-SFT | C-DPO | Rouge-1 | Rouge-2 | Rouge-L | BLEU-4 | Factual | Faithful |
| Qwen | 1.8B | ✓ | | **35.09** | **13.90** | **21.90** | **12.39** | 19.40 | 40.03 |
| | | ✓ | ✓ | 33.78 | 13.37 | 21.66 | 11.88 | **9.02** | **25.55** |
| | 7B | ✓ | | 34.16 | 13.11 | 21.87 | 12.66 | 16.26 | 34.70 |
| | | ✓ | ✓ | **34.99** | **14.13** | **23.11** | **12.93** | **8.33** | **25.41** |
| GPT-4 | 1.8B | ✓ | | 35.68 | 14.66 | 22.31 | 13.36 | 24.32 | 47.13 |
| | | ✓ | ✓ | **35.88** | **14.92** | **22.88** | **13.61** | **16.39** | **33.47** |
| | 7B | ✓ | | 35.72 | 14.45 | 22.91 | 14.17 | 21.17 | 41.26 |
| | | ✓ | ✓ | **36.85** | **15.50** | **24.39** | **14.83** | **12.16** | **30.60** |
| Claude-3 | 1.8B | ✓ | | **34.78** | **14.27** | **21.61** | 10.88 | 22.13 | 39.34 |
| | | ✓ | ✓ | 34.39 | 13.66 | 21.02 | **11.22** | **15.85** | **31.97** |
| | 7B | ✓ | | 34.75 | 13.51 | 21.60 | **11.66** | 20.77 | 37.30 |
| | | ✓ | ✓ | **35.59** | **14.71** | **22.69** | 11.60 | **14.48** | **32.92** |
| Gemini-Pro | 1.8B | ✓ | | 34.57 | 14.28 | 21.47 | 11.85 | 25.68 | 47.54 |
| | | ✓ | ✓ | **35.20** | **14.39** | **22.12** | **11.86** | **17.90** | **38.93** |
| | 7B | ✓ | | 35.20 | 13.99 | 22.35 | 12.70 | 23.63 | 44.13 |
| | | ✓ | ✓ | **36.35** | **15.25** | **23.81** | **13.09** | **15.16** | **37.02** |
| GLM-4 | 1.8B | ✓ | | 37.70 | 16.27 | 24.46 | 12.48 | 10.38 | 18.31 |
| | | ✓ | ✓ | **38.34** | **16.84** | **25.43** | **13.63** | **5.60** | **12.43** |
| | 7B | ✓ | | 38.16 | 16.34 | 25.42 | 13.54 | 10.25 | 22.27 |
| | | ✓ | ✓ | **39.14** | **17.39** | **26.84** | **13.64** | **3.83** | **11.07** |

Additionally, The vanilla SFT method exhibits a significantly higher hallucination rate compared to both TST and SAG methods, possibly because the vanilla SFT primarily relies on summarizations derived from the original articles within the dataset. The summarization process expands the information from summarization to the whole article, which in turn contributes to the higher hallucination rate observed in the vanilla SFT method.

Table 5: Comparison with Vanilla SFT and SAG, Vanilla SFT uses the original summary and style reference as inputs, while SAG utilizes the original summary, style reference, and LLM-generated neutral text as inputs.

| Strategy | Rouge-1 | Rouge-2 | Rouge-L | BLEU-4 | Factual | Faithful |
|---|---|---|---|---|---|---|
| SFT | 34.53 | 13.00 | 21.53 | 12.91 | 35.25 | 65.71 |
| SAG | **36.19** | **14.69** | **23.61** | **14.63** | **14.34** | **34.02** |

Table 6: Comparison with TST and SAG, TST uses LLM-generated neutral text and a style reference as inputs, while SAG additionally includes the original summary as input. Ref. refers to the style reference and Ins. refers to the original summary.

| Method | Ref. | Ins. | Rouge-1 | Rouge-2 | Rouge-L | BLEU-4 | Factual | Faithful |
|---|---|---|---|---|---|---|---|---|
| TST | ✓ | | 34.22 | 13.08 | 21.71 | 13.03 | 21.17 | 41.26 |
| SAG | ✓ | ✓ | **36.19** | **14.69** | **23.61** | **14.63** | **14.34** | **34.02** |

**Effects of Style Reference.** In the previous section, we hypothesized that style can be implicitly defined by the style reference text. In this experiment, we test the extent to which the reference text influences the style of the generated content. We conduct an experiment that removes the style reference for comparison. Additionally, to verify that the style information originates from the same author rather than merely from the same domain, we randomly shuffle the style reference of each

sample in NoteBench, as shown in Tab. 7. Text generated with the inclusion of style reference text achieves significantly higher performance compared to text generated without style reference or with shuffled style references. These results indicate that both strong LLMs and smaller models are capable of learning stylistic information from the reference text.

Table 7: Comparison of different content generation approaches: SLM denotes the Qwen-14B model, Ref. denotes the use of the same user's article as the style reference, and Shuf. denotes the use of a randomly selected user's article as the style reference.

| LLM | SLM | Shuf. | Ref. | Rouge-1 | Rouge-2 | Rouge-L | BLEU-4 |
|---|---|---|---|---|---|---|---|
| Qwen | | | | 29.62 | 11.76 | 19.85 | 10.81 |
| | | ✓ | | 30.45 | 12.09 | 20.16 | 9.49 |
| | | | ✓ | 31.28 | 11.24 | 20.83 | 11.20 |
| | ✓ | | | 33.48 | 12.53 | 21.49 | 12.16 |
| | ✓ | ✓ | | 34.05 | 10.36 | 21.05 | 10.64 |
| | ✓ | | ✓ | **34.50** | **13.34** | **22.44** | **12.96** |
| GPT-4 | | | | 31.11 | 11.02 | 19.72 | 11.14 |
| | | ✓ | | 30.35 | 10.86 | 19.43 | 10.54 |
| | | | ✓ | 35.55 | 14.34 | 23.61 | 14.28 |
| | ✓ | | | 34.16 | 12.68 | 21.25 | 12.60 |
| | ✓ | ✓ | | 33.92 | 9.62 | 19.70 | 11.79 |
| | ✓ | | ✓ | **36.19** | **14.69** | **23.61** | **14.63** |

**Effects of the Stylish Text Filter.** In the training dataset for PersonifyLLM, we use articles posted by the same creator as style reference text. To enhance style relevance, we employ a style classification model to filter out unrelated articles. In this experiment, we evaluate the effectiveness of this style filtering method by also training a model on an unfiltered dataset with identical hyperparameters for comparison. As shown in Table 8, the filtered dataset consistently improves overall performance, validating the effectiveness of the style filter method.

Table 8: The effect of style filter model.

| LLM | Style Filter | Rouge-1 | Rouge-2 | Rouge-L | BLEU-4 |
|---|---|---|---|---|---|
| Qwen | ✗ | 33.33 | 12.47 | 21.30 | 12.52 |
| | ✓ | **34.50** | **13.34** | **22.44** | **12.96** |
| GPT-4 | ✗ | 35.71 | 14.26 | 22.84 | 14.15 |
| | ✓ | **36.19** | **14.69** | **23.61** | **14.63** |

## 5 CONCLUSION

In this paper, we have addressed the challenges of stylish article generation by proposing a novel collaborative training framework that leverages both large language models (LLMs) and small language models (SLMs). Our approach effectively balances style-following capabilities with content generation through a two-stage process of style supervised fine-tuning (S-SFT) and content direct preference optimization (C-DPO). By freezing the LLM to maintain world knowledge and instruction-following capabilities while fine-tuning the SLM, we significantly enhance the stylistic and contextual refinement of generated articles. Our experiments, evaluated on the newly introduced NoteBench benchmark, demonstrate that our method surpasses existing state-of-the-art models in both style consistency and hallucination mitigation. These results affirm the efficacy of our collaborative training framework in generating coherent and contextually relevant stylish content. This work paves the way for further advancements in AI-generated content (AIGC), particularly in refining models' stylistic and content generation capabilities.

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

# A DEMONSTRATIONS

## A.1 AN STYLE IMITATION EXAMPLE

an example of style imitation, including summary, neutral content, style reference and content. The example is translated for reference.

Table 9: Demonstration of the user-generated content along with the generated summary and neutral text. The first line represents the summary, the second line is the neutral text, and the third line is the original text.

| Type | Example | Translated |
|------|---------|------------|
| summary | 今日穿搭推荐：宽松大号黑色T恤搭配侧边较高的短裤，营造男友风的下衣失踪效果。鞋子可选马丁靴或老爹鞋。夏季穿搭要注意防晒。 | Today's Outfit Recommendation: A loose oversized black T-shirt paired with shorts that are higher on the sides, creating a boyfriend-style "pants-disappearing" effect. For shoes, you can opt for Martin boots or dad sneakers. For summer outfits, be sure to pay attention to sun protection. |
| neutral content | 今日穿搭推荐：短裤搭配男友风宽松大T恤。
首先，推荐选择大号的宽松T恤，以黑色为佳，能遮住臀部，营造下衣失踪的效果。
其次，短裤应选择侧边稍高、能视觉拉长腿型的款式，材质可选卫衣料子，但要避免过于花哨。
鞋子方面，可搭配马丁靴或老爹鞋，展现不同风格。
此穿搭适用于夏天，但请注意防晒，防止晒黑。 | Today's Outfit Recommendation: Shorts paired with a boyfriend-style loose T-shirt.
Firstly, choose an oversized loose T-shirt, preferably in black, that can cover the hips to create the "pants-disappearing" effect.
Secondly, for shorts, opt for styles that are slightly higher on the sides to visually elongate the legs. The material can be sweatshirt fabric, but avoid overly fancy designs.
As for shoes, you can pair it with Martin boots or dad sneakers to showcase different styles.
This outfit is suitable for summer, but please remember to use sunscreen to prevent tanning. |
| style reference | 关于缅甸～白色场景更出片啦
要是没有疫情又可以旅游啦
这是20年去缅甸拍的
缅甸不仅是翡翠之城还是佛寺众多的国家
所有到缅甸佛寺观光游览者必须光脚以示敬意
这个寺庙进去就是一排排的白色小塔拍照绝佳
也不能穿露小腿的衣服所以这个连衣裙正好
虽然拍的时候天已经快要黑了但是氛围感十足
对比后面两张阳光明媚的照片也不逊色
苹果原相机拍摄找好角度咔嚓咔嚓就完事
希望疫情快快过去继续周游起来 | About Myanmar  The white scenes are more photogenic now!
If it weren't for the pandemic, we could travel!
These were taken in Myanmar in 2020.
Myanmar is not only the city of jade but also a country with many temples.
All visitors to the temples in Myanmar must go barefoot to show respect.
Entering this temple, there's a row of white stupas—perfect for photos.
No clothes that expose the calves are allowed, so this dress is just right.
Although it was almost dark when I took the photos, the atmosphere is still amazing.
Compared to the two sunny photos behind, it's equally stunning.
Just find the right angle with the iPhone camera, snap, snap, and done!
Hope the pandemic ends soon so we can travel again! |
| content | 全网穿搭逛逛测今日穿搭推荐短裤加男友风大大T恤
1. T恤推荐：
- 夏天终于来啦！虽然偶尔还会有点抽风变冷但是我们可以穿大大的T恤和短裤来一个下衣失踪啦
- T恤就选宽松版大号的就行，争取穿上能把咱的pp盖住就达到效果了
- 黑色更酷
2. 短裤推荐
- 选择侧边稍微高一些的能视觉上拉长我们的大长腿哈哈
- 舒适度的话可以选择那种卫衣料子的短裤当然不要花里胡哨的
3. 鞋子推荐
- 鞋子可以选马丁靴也可以选老爹鞋不一样的风格
总结：这种风格很实用于夏天但是记得涂防晒霜呀，避免晒黑今天穿搭测评就到这里啦 | Check out today's outfit recommendation: Shorts with a boyfriend-style oversized T-shirt
1. T-shirt Recommendation:
- Summer is finally here! Although it can still get a bit breezy and cold occasionally, we can wear oversized T-shirts and shorts for that "pants-disappearing" look.
- Just choose a loose, oversized T-shirt that covers our pp for the perfect effect.
@ Black is cooler
2. Shorts Recommendation:
- Choose shorts that are slightly higher on the sides to visually elongate our long legs, haha.
- For comfort, you can go for sweatshirt-material shorts; just avoid those overly fancy designs.
3. Shoes Recommendation:
- You can go for either Martin boots or dad sneakers for different styles.
Summary:
This style is great for summer, but remember to apply sunscreen to avoid getting tanned. That's the end of today's outfit review. |