# OpenReview forum: "SAG: Style-Aligned Generation via Language Model Collaboration"
_ICLR.cc/2025/Conference — ICLR 2025 Conference Withdrawn Submission_

### Official Review · Reviewer_ZQWM · 2024-10-28

**Soundness:** 3
**Presentation:** 2
**Contribution:** 3
**Rating:** 5
**Confidence:** 4

**Summary:**

The paper introduces a new collaborative training framework that combines the strengths of LLMs and SLMs for generating style articles, outperforming each model individually. Additionally, it establishes a new benchmark, NoteBench, to rigorously assess style-aligned generation.

**Strengths:**

1. The paper presents a new collaborative training framework that combines the strengths of both large and small language models for stylish content generation, leading to better performance than using a single model alone.

2. The paper outlines the style imitation task, enabling a novel approach to generating stylish articles from large unlabeled or semi-labeled datasets. The study also constructed a stylish article dataset with large number of characters, significantly surpasses other datasets used in related models.

3. The proposed method delivers state-of-the-art performance by employing a collaborative training approach for both small and large language models, resulting in notable enhancements in stylish article generation and reduced hallucination.

**Weaknesses:**

1. In the second paragraph of the Introduction part, you'd better provide 2-3 specific examples of recent, influential papers for each approach (training-free and training-based), in order to inform readers about recent research in this field.

2. Too many empty lines in Introduction Section (Line 80 - 84), which makes the paper looks untidy.

3. In Line 86, you mention the difficulties of balancing style-following capabilities with effective content generation. Do you mean the challenge of balancing stylic text generation capabilities and effective model training? There should be a more accurate description here.

4. In the third paragraph of the Introduction part, I think the way of how your framework handles those problems should be emphasized accordingly, in order to clearly show your motivations. Such as describing as "to alleviate catastrophic forgetting of directly fine-tuning LLMs, our framework xxx. To improve generation capabilities of SLMs, our framework xxx."

5. In the Introduction part, people may have different definitions of small language models (SLM). How small a model is that can be named as SLM? Maybe some model names can be listed as examples of the SLM here.

6. In Section 3.1, there should be a formal definition of the style imitation task. You could include key components such as input variables (e.g., content summary, style reference), output variables (e.g., stylized content), and the objective function or evaluation criteria

7. Evaluation Benchmarks of Section 4.1, some recent reference-based scores rather than ROUGE and BLEU should be considered, such as BLEURT (Sellam et al., ACL 2020) and BERT-Score (Tianyi et al., ICLR 2020). On the other hand, you'd better explain why ROUGE and BLEU might be more suitable for evaluating style imitation tasks.

8. In Section 4.2, since there are quite a mount of inferior results in the performance comparison, I think it's better to specifically analyze the reasons of leading to the performance reduction here.

**Questions:**

1.  In Section 3.2, When you conduct the data curation stage, what are those original articles? How you collected them? What they look like?

2. In Section 3.3 (Line 247-248), can you explain how you SLM stage replicate tone, vocabulary choice, and sentence structure in details? Since you mentioned these stylish traits, you should explain each of them accordingly.

3. Where is the detailed description of your proposed NoteBench benchmark? Since you particularly listed it as one of your main contributions, I think there should be a specific part from your main content of illustrating NoteBench, and for instance, including following points:

* The composition of NoteBench (e.g., number and types of samples)
* How it differs from existing style transfer benchmarks
* The evaluation criteria specific to NoteBench
* How NoteBench addresses the unique challenges of style imitation tasks

---

### Official Review · Reviewer_AW9n · 2024-10-29

**Soundness:** 2
**Presentation:** 2
**Contribution:** 2
**Rating:** 5
**Confidence:** 4

**Summary:**

This paper introduces a novel method called Style-Aligned Article Generation (SAG) that leverages the strengths of both large language models (LLMs) and small language models (SLMs) to generate stylish, contextually accurate articles. The approach addresses the limitations of individual models by freezing LLMs to maintain their instruction-following capabilities, while fine-tuning SLMs using style-specific data to enhance the stylistic aspects of the output. The authors propose a collaborative training framework involving two stages: Style Supervised Fine-Tuning (S-SFT) and Content Direct Preference Optimization (C-DPO) to achieve superior performance in style imitation, compared to models like GPT-4. A new benchmark, NoteBench, is introduced to evaluate style consistency and hallucination rates, showing that this method outperforms state-of-the-art models in both areas. The paper also provides an extensive experimental setup and analysis to validate its findings.

**Strengths:**

1. A framework has been proposed, which brings consistent performance improvements across multiple models.
2. A style dataset and a benchmark for evaluation have been provided.

**Weaknesses:**

1. The collaborative training framework introduced by the authors is based on using LLM to generate data for training SLM. However, whether using prompting to guide LLM or the two-stage training of SLM through SFT-DPO is quite common, and this approach seems to lack technical innovation.

2. Since the task of this paper focuses on style, using BLEU and ROUGE as evaluation metrics appears somewhat inadequate. It would be more appropriate to introduce metrics that assess the style, or consider incorporating human evaluation.

3. Although the authors conducted extensive experiments to demonstrate the effectiveness of the framework in improving both SLM and LLM independently, they did not compare their method with other Style Transfer approaches, which makes the framework's effectiveness less convincing.

**Questions:**

1. In the SFT stage, what data is used to supervise the training process? If it is the user's original content, how do you ensure that the style of the style reference article remains consistent with it? If not, could you please clarify what is used?

2. The explanation of the DPO stage seems too vague. Could you please provide a more detailed explanation, especially regarding: 1) How is the LLM used to correct the hallucinated segments of the generated content? 2) How do you instruct the LLM to incorporate additional information from the reference content?

3. We noticed that in Table 3, the improvements for GPT-4 and GLM-4 are significantly smaller compared to the other models. Could you provide an explanation for this?

---

### Official Review · Reviewer_m9Nc · 2024-11-01

**Soundness:** 2
**Presentation:** 2
**Contribution:** 2
**Rating:** 3
**Confidence:** 5

**Summary:**

The paper presents a novel collaborative training framework for style-aligned article generation, leveraging both large language models and small language models. The authors propose freezing the LLM to maintain its robust instruction-following capabilities while fine-tuning the SLM on style-specific data. The paper introduces a new benchmark, NoteBench, to evaluate style consistency and factual accuracy, claiming significant improvements in style alignment compared to GPT-4 and other baseline models. The method also reduces hallucination rates, addressing key challenges in stylish content generation.

**Strengths:**

The paper introduces an interesting method by utilizing LLMs to construct training data through a “reverse” generation process, which helps expand the available data for training. This approach allows for more diverse data generation without relying solely on limited, manually curated datasets. By generating neutral content and then applying stylistic adaptation, the framework creatively expands the training pool, which could enhance model training in low-resource scenarios.

**Weaknesses:**

The most significant issue in the paper is the inconsistency in the reported results across different tables.

- Table 3 vs. Table 4: Taking the results of the 1.8B models as an example, the Style Consistency score for Qwen in Table 3 is identical to the score for Gemini-Pro in Table 4, and vice versa. I believe one of the tables has swapped the results for these two models, which undermines the reliability of the findings.
- Table 6 vs. Table 7: A more serious issue arises when comparing results under different LLMs in Table 6 and Table 7. According to line 371, Table 6 uses Qwen-72B-Chat as the LLM for reference generation, where the Rouge-1 score for SAG is 36.19. However, in Table 7, where GPT-4 is the LLM (line 454), the SAG model achieves the same Rouge-1 score of 36.19, and all other metrics are identical. This suggests a significant error in reporting, as these results should not be identical under different LLMs.

Due to these major errors in the results, I think the paper’s claims are unreliable. The authors need to thoroughly check and correct the results in the tables to ensure they align with the actual experimental findings.

Beyond these issues, there are a few additional concerns:

1. Inconsistent experimental settings. The experimental setups vary across different sections of the paper, making it difficult to compare results between experiments. For example, the main experiments in Table 4 use Qwen 1.8B and Qwen 7B as the SLMs, but Table 5 and 6 do not clearly specify the SLM used, while Table 7 employs Qwen 14B as the SLM. These inconsistencies in the experimental settings cast doubt on whether the conclusions drawn from the analysis experiments can be applied to the main experiments. The authors should standardize the experimental settings across all experiments unless there is a specific reason to vary them, which should be clearly explained.
2. Lack of comparison with existing methods. The paper evaluates its performance solely on its self-created dataset and does not provide comparisons with existing methods. Given the number of well-established datasets and methods for style transfer and role-playing, the lack of comparisons makes it difficult to assess the effectiveness of the proposed approach. Including comparisons with existing approaches would strengthen the paper’s contribution and validate its claims.
3. Unclear implementation of baselines. The paper lacks details on how the baseline models were tested in Table1, including whether zero-shot, few-shot, or other techniques were used. Providing this information, along with example prompts in the appendix, would help readers evaluate the baselines properly.
4. Lastly, the paper claims in line 224 that “The detailed prompts and examples are provided in Appendix A,” but only examples are included, with no prompts.

**Questions:**

Please refer to and address the issues mentioned in the weaknesses section. In particular, there are inconsistencies in the experimental results and unclear experimental settings across different tables. Additionally, the paper would benefit from comparisons with more benchmarks and baselines beyond the self-created dataset to better assess the effectiveness of the proposed method in relation to existing approaches. Clarifying and resolving these points would significantly improve the paper’s clarity and reliability.

---

### Official Review · Reviewer_e1Zc · 2024-11-05

**Soundness:** 1
**Presentation:** 2
**Contribution:** 2
**Rating:** 3
**Confidence:** 4

**Summary:**

This paper introduces a collaborative training framework designed to enhance stylish content generation by combining the strengths of both large and small language models (LLMs and SLMs). The framework consists of two training stages: Style-Supervised Fine-Tuning (S-SFT) and Content Direct Preference Optimization (C-DPO).

The authors also introduce NoteBench, a benchmark to evaluate style consistency and factual accuracy in generated content. Experimental results show that the SAG framework surpasses state-of-the-art performance, achieving notable improvements in style alignment metrics (such as ROUGE and BLEU scores) and effectively reducing hallucination rates, outperforming standalone LLMs.

**Strengths:**

- The paper introduces a unique framework combining large language models (LLMs) with small language models (SLMs) for style-specific content generation, leveraging each model’s strengths.
- The NoteBench benchmark could be useful for future studies on style imitation.

**Weaknesses:**

- The paper provides strong motivation for the collaborative training framework; however, it lacks clarity on the necessity of introducing a new task to demonstrate its effectiveness. Specifically, the proposed task—imitating a specific style while generating content from a summary—appears somewhat impractical and lacks clear theoretical or practical justification. The value of this task remains ambiguous, as the paper does not convincingly demonstrate its relevance or potential applications in real-world scenarios.
- The fatal weakness of this paper is its reliance on ROUGE and BLEU scores within the NoteBench benchmark to assess style consistency. While these metrics are frequently used for measuring lexical overlap, they fall short in capturing stylistic nuances such as tone, vocabulary, and structure, which are central to the style imitation task proposed by the paper. Let alone that they're indeed not reliable evaluation metrics for NLG tasks.
- btw. This is the first time I see someone use metrics like BLEU to evaluate the style. Too often, they were used to measure content preservation in style transfer tasks.
- Similar problems apply to the evaluation of hallucination. The paper relies on LLM-based annotation for hallucination detection without establishing the reliability of this approach. Emerging research, such as the hallucination detection task in Semeval 2024, suggests that LLMs are not yet fully dependable for hallucination annotation (see: https://helsinki-nlp.github.io/shroom/2024).

**Questions:**

See my comments in weaknesses.

---

### Note · Authors · 2024-11-18

I have read and agree with the venue's withdrawal policy on behalf of myself and my co-authors.